# The Role of mpMRI in the Assessment of Prostate Cancer Recurrence Using the PI-RR System: Diagnostic Accuracy and Interobserver Agreement in Readers with Different Expertise

**DOI:** 10.3390/diagnostics13030387

**Published:** 2023-01-20

**Authors:** Chiara Bergaglio, Veronica Giasotto, Michela Marcenaro, Salvina Barra, Marianna Turazzi, Matteo Bauckneht, Alessandro Casaleggio, Francesca Sciabà, Carlo Terrone, Guglielmo Mantica, Marco Borghesi, Alessio Signori, Bruno Spina, Nataniele Piol, Elisa Zanardi, Giuseppe Fornarini, Jeries Paolo Zawaideh

**Affiliations:** 1Department of Health Sciences (DISSAL), University of Genova, 16126 Genova, Italy; 2Department of Radiology, IRCCS Ospedale Policlinico San Martino, 16132 Genova, Italy; 3Department of Radiation Oncology, IRCCS Ospedale Policlinico San Martino, 16132 Genova, Italy; 4Nuclear Medicine Unit, IRCCS Ospedale Policlinico San Martino, 16132 Genova, Italy; 5Department of Urology, IRCCS Ospedale Policlinico San Martino, 16132 Genova, Italy; 6Department of Surgical Integrated Sciences (DISC), Urology Section, University of Genova, 16126 Genova, Italy; 7Pathology Unit, IRCCS Ospedale Policlinico San Martino, 16132 Genova, Italy; 8Anatomia Patologica Universitaria Unit, IRCCS Ospedale Policlinico San Martino, 16132 Genova, Italy; 9Academic Unit of Medical Oncology, IRCCS Ospedale Policlinico San Martino, 16132 Genova, Italy; 10Medical Oncology Unit 1, IRCCS Ospedale Policlinico San Martino, 16132 Genova, Italy

**Keywords:** prostate cancer, multiparametric MRI, biochemical recurrence

## Abstract

Background: treated prostate cancer (PCa) patients develop biochemical recurrence (BCR) in 27–53% of cases; the role of MRI in this setting is still controversial. In 2021 a panel of experts proposed a “Prostate Imaging-Recurrence Reporting” (PI-RR) score, aiming to standardize the reporting. The aim of our study is to evaluate the reproducibility of the PI-RR scoring system among readers with different expertise. Methods: in this monocentric, retrospective observational study, the images of patients who underwent MRI with BCR from January 2017 to January 2022 were analyzed by two radiologists and a radiology resident. Sensitivity, specificity, positive predictive value (PPV), negative predictive value (NPV), and accuracy were obtained. Interobserver agreement was calculated. The percentage of the PI-RR score of 3 was estimated to find out the proportion of uncertain exams reported among the readers. Results: a total of seventy-six patients were included in our study: eight previously treated with RT and sixty-eight who underwent surgery. The accuracy range was 75–80%, the sensitivity 68.4–71.1%, the specificity 81.6–89.5%, PPV 78.8–87.1%, and NPV 72.1–75.6%. The inter-reader agreement using a binary evaluation (PI-RR ≥ 3 as positive mpMRI) demonstrated a correlation coefficient (k) of 0.74 (95% CI: 0.62–0.87). The percentage for the PI-RR score of 3 was 6.6% for reader one, 14.5% for reader two, and 2.6% for reader three. Conclusion: this study confirmed the good accuracy of mpMRI in the detection of local recurrence of PCa and the good reproducibility of PI-RR score among all readers, confirming it to be a promising tool for the standardization of the assessment of patients with BCR.

## 1. Introduction

Despite advances in prostate cancer (PCa) primary intended curative therapy, 27–53% of patients develop biochemical recurrence (BCR), with higher percentages depending on the preoperative risk and stage of cancer [1,2].

The pattern of PSA rise has been included into clinical nomograms and helps clinicians to predict whether the PCa recurrence is more likely to be local or systemic; however, only imaging techniques can localize it with certainty [3].

The role of multiparametric MRI in the evaluation of the prostatic bed in patients with biochemical recurrence (BCR) is controversial. The European Association of Urology (EAU) guidelines recommends Positron Emission Tomography/Computed Tomography (PET/CT) imaging, preferably with radiolabeled Prostate-Specific Membrane Antigen (PSMA) in patients with BCR after radical prostatectomy (RP) and an MRI in patients with BCR after radiotherapy (RT) [4]. 

In 2021, a panel of experts published the International-Consensus Guidelines for Prostate Imaging-Recurrence Reporting (PI-RR), aiming to standardize image acquisition and reporting of local recurrence in patients with BCR [5]. 

However, this score has been validated using only single-center studies, with cases evaluated by expert radiologists requiring further validation [6,7].

In this light, the primary aim of our study is to evaluate the reproducibility of the PI-RR scoring system by considering the diagnostic accuracy of MRI in the detection of local PCa recurrence and the interobserver agreement among readers with different expertise. The secondary purpose is to identify a correlation between clinical risk factors and PCa recurrence. 

## 2. Materials and Methods

### 2.1. Patient Sample and Study Design

This is a monocentric, retrospective observational study funded by Grant for Current Research R779A with the allocation of resources by the Italian Ministry of Health.

The patients referred to our Institution with biochemical recurrence from January 2017 to January 2022, previously treated with radical prostatectomy (RP) or Radiant Therapy (RT), underwent a mpMRI to rule out a local recurrence, according to the EAU Guidelines on prostate cancer criteria. 

The definition of biochemical recurrence identifies the cut-off in the value of 0.2 ng/mL after radical prostatectomy [8]; while after radiotherapy, the “Phoenix” definition used identifies biochemical recurrence at any value ≥ 2 ng/mL compared to the post-radiotherapy nadir [9].

### 2.2. Exclusion Criteria

The exclusion criteria used to select patients for this study were: (a) patients who already received androgen-deprivation therapy before undergoing the MRI examination; (b) the presence of magnetic susceptibility artefacts which degrade the quality of DWI sequences; and (c) patients who did not attend the follow-up visits at our institution. 

### 2.3. MRI Acquisition

The mp-MRI was performed using a 1.5 T MRI (Magnetom AERA, Siemens, Munich, Germany), the MRI protocol was compliant with the recommendation by PI-RADS v2 Guidelines and updated to PI-RADS v2.1 Guidelines after 2019 [10].

T2w images were obtained in the axial, coronal, and sagittal plane; Diffusion-weighted imaging (DWI) and Dynamic contrast enhancement (DCE) were obtained with the same slice thickness and plane as the T2w axial sequence, in order to obtain a match. 

DWI images were obtained through single-shot echo-planar sequence with a high b-value (1400 s/mm^2^) and another sequence with 3 different b-values (0, 750 and 1000 s/mm^2^); this latter was used to obtain the apparent diffusion coefficient (ADC) map.

DCE acquisition was obtained during the intravenous injection of a gadolinium-based contrast agent with 0.2 mL/kg dose at a flow rate of 3 mL/s followed by 15 mL of saline solution; the temporal resolution was 9 s. Full protocol parameters that can be found in Appendix A.

### 2.4. MRI Analysis

Image analysis was performed independently using a radiology resident (Reader 1) and by two radiology consultants (Reader 2 and 3), respectively with 3, 6, and 10 years of experience in prostate MRI.

Each radiologist evaluated each sequence (DCE, DWI and T2) using the PI-RR criteria and assigned a final score for each patient accordingly to the PI-RR algorithm; the clinical and histological data of the primitive tumor as well as the side were known for each patient. If there were two suspected areas, only the lesion with the higher PI-RR score was considered. Figure 1. Case examples of each PI-RR overall score

Axial MRI images of patients previously treated with radical prostatectomy.

PI-RR 1: 81-year-old man with prior 4 + 3 Gleason PCa, (prostate-specific antigen [PSA] level = 0.27 ng/mL). Symmetric and hypointense vesicourethral anastomosis, without restriction neither early contrast enhancement.

PI-RR 2: 78-year-old man with prior 3 + 4 Gleason PCa, (prostate-specific antigen [PSA] level = 0.34 ng/mL). Hypointense T2w vesicourethral anastomosis, without restriction neither early contrast enhancement, however with asymmetrical thickness on the left vesicourethral anastomosis (arrowhead).

PI-RR 3: 74-year-old man with prior 4 + 5 Gleason PCa, (prostate-specific antigen [PSA] level = 0.58 ng/mL). Hypointense T2w vesicourethral anastomosis, with posterior asymmetrical thickness (at 6 o’ clock), with focal mid–late enhancement on the DCE image (arrow).

PI-RR 4: 84-year-old man with prior 4 + 3 Gleason PCa, (prostate-specific antigen [PSA] level = 5.5 ng/mL). A mass-like iso-hypointense lesion at T2-weighted imaging (at 6–9 o’clock), with focal marked hyperintensity at high b-value DWI (arrow) and focal early enhancement on the DCE image (arrowhead); primary cancer was however contralateral (located on the posteromedial left peripheral zone). 

PI-RR 5: 77-year-old man with prior 3 + 4 Gleason PCa, (prostate-specific antigen [PSA] level = 5.94 ng/mL). A mass-like iso-hypointense lesion at T2-weighted imaging (at 4–6 o’clock), with focal marked hyperintensity at high b-value DWI (arrow) and focal early enhancement on the DCE image (arrowhead) on the same side of the primary cancer.

### 2.5. Reference Standard Definition

Results were validated using a composite reference standard consisting of heterogeneous parameters. The criteria to assess a true positive recurrence on (a) the confirmation of the positivity of the lesion using correlative imaging (i.e., [18F]F-Fluorocholine or [18F]F-PSMA-1007 PET/CT); (b) the reduction of PSA levels after salvage RT. The criteria to assess a true negative are based on: (a) a negative choline or PSMA PET/CT; (b) a positive choline or PSMA PET/CT at a site different from the prostatic bed (osseous of lymph-nodal metastases), whose targeted salvage treatment led to a PSA reduction; (c) the appearance over time of a positive target in an imaging investigation performed as a follow up, not visible even retrospectively on MRI; (d) “subsequent spontaneous PSA decrease without any therapy”.

### 2.6. Statistical Analysis

General characteristics of mean, median, and interquartile range (IQR) were used to summarize continuous variables, with counts and percentages used to summarize categorical data. 

Sensitivity, specificity, positive predictive value (PPV), negative predictive value (NPV), and accuracy were obtained considering a PI-RR score < 2 associated with a negative outcome and a PI-RR score ≥ 3 as a positive mpMRI.

The percentage of PI-RR score 3 was estimated, to find out the proportion of uncertain exams reported among the three readers.

Inter-reader agreement was evaluated using the Gwet’s kappa statistical analysis for each PI-RR score and for the binary evaluation (of positive and negative MRI). Inter-reader agreement was calculated for each sequence (DCE, DWI, T2) and for final PI-RR score assessment.

Differences between groups were analyzed using the Mann–Whitney test. *p* values < 0.05 were considered to be statistically significant. 

Univariate and multivariable logistic-regression analyses were performed to evaluate the correlation between clinical risk factors and the local PCa recurrence. For the multivariable model only the significant variables at the univariate analyses were selected. All variables with a *p*-value > 0.10 in the multivariable model were then removed for the final model.

All statistical analyses were performed using Stata (v.16; StataCorp) [11].

## 3. Results

In this study, 108 consecutive patients underwent mpMRI at our institution for biochemical recurrence; of those thirty-two patients were excluded (details can be found in Figure 2) with seventy-six patients finally included in our study, eight who were previously treated with RT and sixty-eight who previously underwent surgery.

### 3.1. Demographic Characteristics 

The mean age of patients was 72.4 years, with a median of 73.4 (UP 76.1 and LQ 69.1). The group of patients who underwent RT had a mean age of 77.1 years, with a median age of 78.6 (UP 81.1 and LQ 74.5), while the group of patients who underwent RP had a mean age of 71.7 years, with a median age of 73.3 (UP 75.3 and LQ 68.9).

The mean disease-free interval was 5.9 years, with a median of 5.1 (UP 8.5 and LQ 1.5); among the group of patients who underwent RT, the mean disease-free interval was 9.3 years, with a median of 8.7 years (UP 11.4 and LQ 8); for the group of patients who underwent RP, the mean disease-free interval was 5.5 years, with a median of 4.7 years (UP 8.5 and LQ 1.3).

The mean PSA value at the time of the MRI examination was 1.01 μg/mL, with a median of 0.27 μg/mL (UP 0.76 and LQ 0.21 μg/mL); the group of patients who underwent RT had a mean PSA value of 1.8 μg/mL, with a median of 1.6 μg/mL (UP 2.2 and LQ 1.1 μg/mL); the group of patients who underwent RP had a mean PSA value of 0.95 μg/mL, with a median of 0.26 μg/mL (UP 0.59 and LQ 0.20 μg/mL). 

Among the 76 patients included in the study, 38 patients had recurrent disease in the prostatic bed while in the other 38 were negative for local disease; results of recurrent disease were defined accordingly to composite reference standard consisting of heterogeneous parameters that can be found in Appendix A. 

### 3.2. MRI Detection Performance

MpMRI detection performance using a binary evaluation (considering a PI-RR score ≥ 3 as positive) showed a sensitivity in identifying recurrence in the prostatic bed of 68.4–71.1%; specificity of 81.6–89.5%; accuracy of 75–80%; Positive predictive value range (PPV) of 78.8–87.1% and Negative predictive value (NPV) of 72.1–75.6%. MpMRI performances for each reader are reported in Table 1.

The percentage of a PI-RR score of 3 (score of uncertainty) was 6.6% for reader one, 14.5% for reader two, and 2.6% for reader three.

Diagnostic performance for each reader for recurrence detection without a score of uncertainty (Pi-RR score of 3) can be found in Appendix A.

The inter-reader agreement using a binary evaluation demonstrated an overall correlation coefficient (k) of 0.74 (95% CI: 0.62–0.87) and, divided into sequences, of 0.72 (95% CI 0.59–0.85) for DWI sequence, of 0.73 (95% CI 0.61–0.86) for DCE sequence, and 0.75 (95% CI 0.63–0.87) for T2 sequence. 

In the group of patients who underwent radical prostatectomy, 83% performed the MRI with a PSA < 1 ng/mL, while the remaining 17% with a PSA ≥ 1 ng/mL.

In the “low” PSA group, the percentage of patients with a positive outcome (using the above-mentioned composite reference standard) were 42.5% with disease with an average size of 4 mm, while in patients with “high” PSA, the percentage of positives was 77.8% with an average size of 8 mm. 

MpMRI detection performance using a binary evaluation (positive PI-RR score ≥ 3) showed a higher accuracy in the “high” PSA group ranging between 79–86% versus 66–72% in the “low” PSA group. A complete diagnostic performance for each reader in the different PSA setting can be found in Appendix A.

Inter-reader agreement analysis according to Gwet’s kappa calculated for individual score demonstrated that the overall correlation coefficient interclasses is 0.62 (95% CI: 0.51–0.73); dividing it by sequences the overall correlation coefficient for the DWI sequence is 0.60 (95% CI: 0.49–0.71), for the DCE 0.63 (95% CI: 0.52–0.75), and for the T2 sequence 0.57 (95% CI: 0.46–0.69). 

### 3.3. Correlation Analysis between Local Recurrence and Risk Factors

The univariate analysis shows the statistically significant correlation of the PI-RR ≥ 3 score (with a *p* value *p* < 0.001) and local recurrence (reader one: OR: 9.59 (CI 3.30–27.92); reader two: OR 20.86 (CI 5.97–72.89) and reader three: OR 16.20 (5.01–52.36); this result is confirmed using the multivariate analysis (OR 7.72 (2.17–27.44); *p* = 0.002, statistically significant for all the readers regardless of the expertise. 

The only other PSA values at BCR shows a OR 2.69 (CI 1.05–6.92), with a *p* value = 0.04 in the univariate analysis, but this result is not confirmed using the multivariate analysis (OR 2.22 and *p* = 0.09), neither in the single group analysis based on treatment: OR 2.25 (CI 0.88–5.72); *p* = 0.089. These results are shown in Table 2.

## 4. Discussion

### 4.1. Accuracy

This study confirms that mpMRI plays a relevant role in the detection of pathologic tissue in patients with BCR, who underwent RP and RT, as it is described in the literature [12,13].

These results are in accordance with a recent systematic review [14] that confirmed the performance of mpMRI in the detection of local recurrence, with a median sensitivity of 84% and a median specificity of 92% for the DCE sequences combined with T2 sequences; DWI sequences, combined with T2 sequences, show a median sensitivity of 93% and a median specificity of 91%. 

Furthermore, our results are consistent with more recent studies that used the PI-RR score: Pecoraro et al. presented similar values for patients who underwent RT; they reported a sensitivity of 71–81%, a specificity of 74–93%, and an accuracy 77–92%, in the RT group a sensitivity of 59–83%, specificity of 87–100% and an accuracy of 80–88%; while Ciccarese et al. had a similar sensitivity of 84.6%, with a lower specificity of 33.3% and an accuracy of 68.4%, this may be explained since they only used biopsy-proven lesions [6,7]. 

PSA values are expected to be representative of the volume of the recurrent disease, thus a higher PSA value is likely to represent a bigger recurrence volume. As also observed by Ciccarese et al., the PI-RR in the scenario of low PSA may have lower diagnostic accuracy and a negative Pi-RR may indicate a lower relapse volume, not detectable in mpMRI [7].

Our results pointed out that the more-expert readers show a higher accuracy, but this was not statistically higher compared to resident results (reader 1). Moreover, at the univariate and multivariate analysis, a PI-RR ≥ 3 score predicted the PCa local recurrence, regardless of the previous readers expertise. Therefore, the PI-RR score demonstrates the reproducibility even if it is used by a less-expert reader. 

On the other hand, it is important to notice that experience plays a fundamental role of diagnosis confidence, so much in fact that among the two more-expert readers (reader two and three) who obtained similar accuracy values, the less-experienced one had a higher call of uncertainty of a PI-RR score of 3. 

Predictive risk factors such as PSA value, positive margin (R1), and PCa stage and Gleason score were not statistically predictive of prostatic bed disease in patients with biochemical recurrence; this result was reported also by Pecoraro et al. [6].

### 4.2. Agreement 

One of the main aims of PI-RR was to standardize MRI reporting. In our analysis, the PI-RR score shows a good reproducibility among readers with a correlation coefficient of k = 0.74 (CI: 0.62–0.87). This result is inferior compared with the correlation coefficient presented by Pecoraro et al. [6] (k = 0.87 interclasses) and Ciccarese et al. [7] (K = 0.884, *p* < 0.001), but it is however acceptable considering the marked difference of experience among readers.

The analysis of the inter-reader agreement binarized for PI-RR score shows substantially stackable values for the single sequence (DWI 72%, DCE 73%, T2 75%), while the inter-reader agreement analyzed as a continue-variable DCE sequence has the highest agreement (DWI: 60%. DCE: 63% T2: 57%). 

DCE was reported to have the higher agreement also by Rouvière et al., who obtained a k of 0.63–0.70 for dynamic contrast-enhanced images compared to k 0.18–0.39 for T2-weighted images in the post-RT setting [15]. This result was expected since DCE is the “dominant” sequence used to evaluate the local recurrence after radical prostatectomy. 

### 4.3. PET/CT

Even though the aim of this study was not to compare mpMRI and PET-CT, in our cohort mpMRI demonstrates as not inferior to PET/CT in the evaluation of the local recurrence; in fact, no negative cases at the MRI evaluation have subsequently been considered positive PET-CT.

These results are in accordance with the literature, stating that the MRI showed the same—or a higher—accuracy of PSMA-PET/CT in the evaluation of prostatic bed [16]. 

Ciccarese et al., using PI-RR score, obtained a higher detection rate compared with both choline-PET/CT and PSMA-PET/CT for local recurrence, [7] while Radzina et al. identified a superiority of PSMA-PET/CT only for the detection of distant and lymph node metastases [17]; as well, in our study eleven patients had a positive PET-CT at another location (one patient had a bone metastasis and ten patients had lymph nodal metastases).

In their editorial, Mottaghy et al. highlight that the two imaging modalities are complementary rather than competing [18], and the joint use of the two techniques has been reported to improve accuracy [19]. According to these promising data, recently Panebianco et al. [20] proposed a diagnostic flowchart to evaluate patients with BCR based on risk factors to target patients’ management as much as possible. 

### 4.4. Pitfalls

In our experience we observed three drawbacks of the current PI-RR classification; the first one is the absence of a dimensional cut-off; a size criterion may be able to increase standardization, but the fibrotic reaction is extensive by definition and a dimensional cut-off could lead to misinterpretation and an increase in false-positive exams [21].

Secondly, while there is no doubt that the knowledge of the side of the primary lesion is fundamental in patients who underwent radiotherapy, this may become less important in patients who underwent radical prostatectomy, where the most common sites of recurrence have been described to be the vesicourethral anastomosis, seminal vesicles remnants, and the base of the bladder [22,23].

The third one is that PI-RR does not consider the presence of a remnant of benign prostatic adenoma growing in the prostatic bed; in this case, the T2 sequences are fundamental in differentiating benign tissue from local recurrence [24] (Figure 3).

Shown are bright heterogenous tissue consisting with normal adenoma; however, this area shows moderate restriction and early contrast enhancement on DCE would have been classified as PI-RR five, since DCE is the “dominant” sequence.

The images of this patient belong to a “study case” who was acquired with endorectal coil in our institution earlier than the period of study and not included in the analysis.

### 4.5. Limitations

Our study has some limitations: the retrospective nature of this study led to a little cohort of patients, mainly consisting of patients treated with radical prostatectomy; this limitation likely reflects a selection bias in identifying patients with BCR; this limitation is a common drawback with other PI-RR studies, in particular Pecoraro et al., who also had a slight majority of patients who underwent RP while Ciccarese et al. analyzed only surgical patients [6,7].

Additionally, patients who underwent prior hormonal therapy were not included, as this therapy could compromise the visualization of the recurrence on MRI.

For the above limitations the results cannot be interpreted as conclusive and may not represent all the general population, especially concerning patients who underwent RT.

A further limitation is the single-center design of this study; validation of the PI-RR score using MRI images from multiple centers to assess the reproducibility of this score remains an unmet need.

Another limitation is that in our center, according to EAU 2022 Guidelines [4], patients with BCR are offered salvage radiotherapy on the prostatic bed if PSA values are high in two consecutive measurements, even though PET/CT is negative or not performed. This may lead to an overestimation of false-negative patients. 

Finally, given the retrospective nature of the study, PET/CT imaging with two different radiotracers has been used to define the reference standard. This might have influenced the obtained results since [18F]F-Fluorocholine PET/CT might have been less accurate than [18F]F-PSMA-1007 PET/CT [25], especially at low PSA levels, possibly confirming it as a true negative which could have been a false negative in MRI. Further prospective studies are needed to overcome this limitation. 

## 5. Conclusions

This study confirmed the good accuracy of mpMRI in the detection of local recurrence of prostate cancer and the good reproducibility of the PI-RR score among all readers, confirming it to be a promising tool for the standardization the assessment of patients with BCR.

These data have to be confirmed by larger and prospective studies but suggest that in the future a tailored approach to imaging, based on the risks of patients, can be proposed.

## Figures and Tables

**Figure 1 diagnostics-13-00387-f001:**
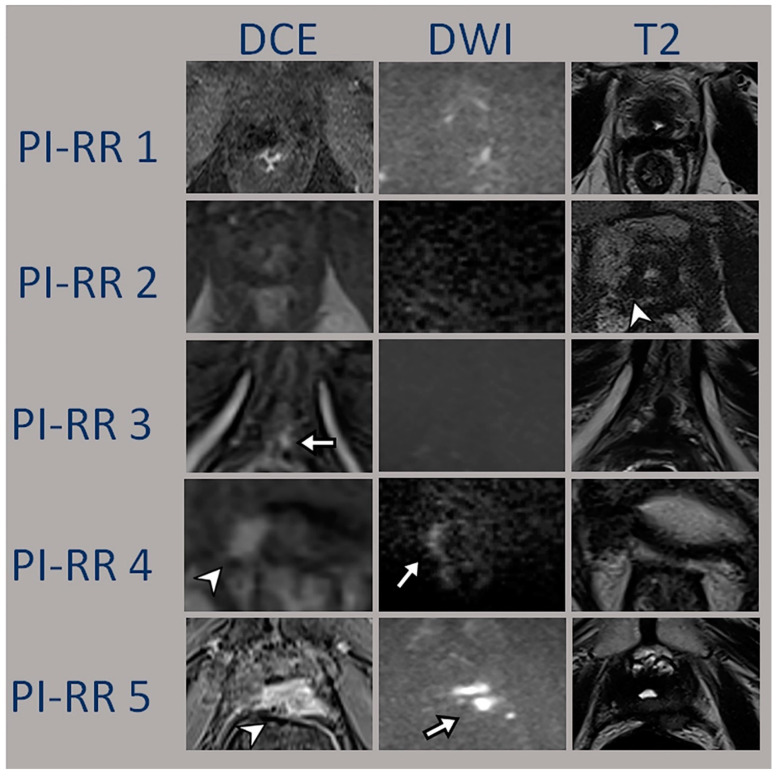
Examples of mp MRI in patients previously treated with radical prostatectomy.

**Figure 2 diagnostics-13-00387-f002:**
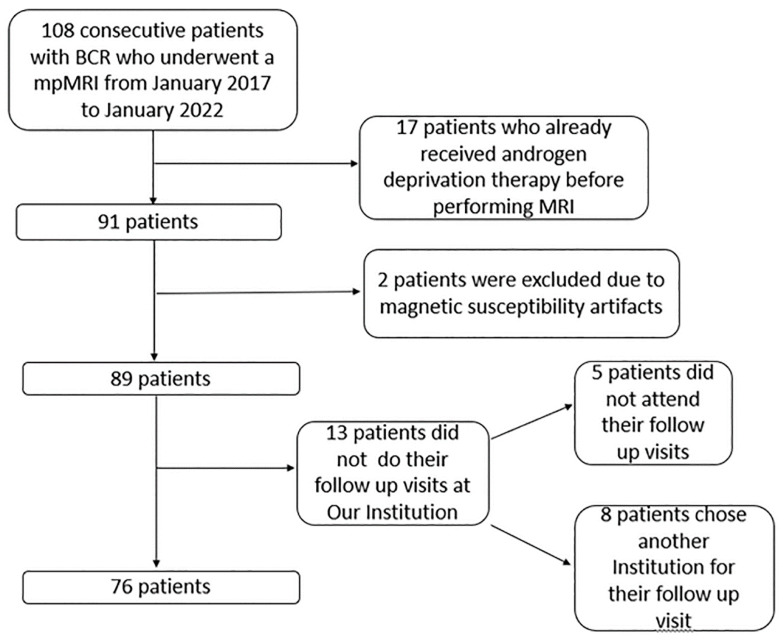
Exclusion criteria flowchart with reasons and excluded numbers.

**Figure 3 diagnostics-13-00387-f003:**
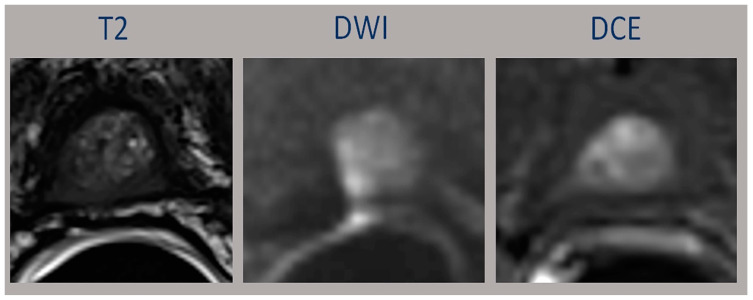
Prostatic adenoma growing in the prostatic bed.

**Table 1 diagnostics-13-00387-t001:** Diagnostic performance for each reader for recurrence detection.

	Reader 1	Reader 2	Reader 3
Sens	68.4 (51.3–82.5)	71.1 (54.1–84.6)	71.1 (54.1–84.6)
Spec	81.6 (65.7–92.3)	89.5 (75.2–97.1)	86.8 (71.9–95.6)
AUC	0.75 (0.65–0.85)	0.80 (0.71–0.89)	0.79 (0.70–0.88)
PPV	78.8 (61.1–91)	87.1 (70.2–96.4)	84.4 (67.2–94.7)
NPV	72.1 (56.3–84.7)	75.6 (60.5–87.1)	75 (59.7–86.8)

**Table 2 diagnostics-13-00387-t002:** Univariate and multivariate correlation analysis.

	Univariable OR (95% CI); *p*-Value	Multivariable ^ OR (95% CI); *p*-Value
Age (1-year)	1.10 (1.01–1.19); *p* = 0.026	
Gleason (1-point)	0.67 (0.39–1.14); *p* = 0.14	
Stage of disease		
<T3	1.00 (ref)	
≥T3	0.59 (0.24–1.46); *p* = 0.25	
Therapy		
PR	1.00 (ref)	
RT	8.35 (0.97–71.65); *p* = 0.053	
R		
0	1.00 (ref)	
1	1.23 (0.32–4.72); *p* = 0.76	
PSA	2.69 (1.05–6.92); *p* = 0.040	2.22 (0.88–5.53); *p* = 0.088
PSA in RP ^§^	2.25 (0.88–5.72); *p* = 0.089	
Score		
<3	1.00 (ref)	1.00 (ref)
≥3 (Reader 1)	9.59 (3.30–27.92); *p* < 0.001	
≥3 (Reader 2)	20.86 (5.97–72.89); *p* < 0.001	
≥3 (Reader 3)	16.20 (5.01–52.36); *p* < 0.001	7.72 (2.17–27.44); *p* = 0.002 *

^ Stepwise selection including only those significant at univariable analysis and with a *p* < 0.10 after inclusion in the multivariable model. * Only the reader with most experience was considered in the multivariable model. Results were, however, confirmed also with the inclusion of score results from reader 1 (*p* = 0.007) or reader 2 (*p* = 0.001). ^§^ Only PR were included in this variable. RT group size was insufficient for robust univariate analysis.

## Data Availability

The data that support the findings of this study are available on request from the corresponding author.

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
