# Peer review of "The Role of mpMRI in the Assessment of Prostate Cancer Recurrence Using the PI-RR System: Diagnostic Accuracy and Interobserver Agreement in Readers with Different Expertise"

_diagnostics, 2023, doi:10.3390/diagnostics13030387_

Round 1

Reviewer 1 Report

This is an interesting manuscript dealing with a timely topic. The research question is clear and relevant. The study design is straightforward and properly described. I have only some minor comments:

1) It has been shown that, in the setting of BCR, when PSA level is below 1 ng/ml the accuracy of MRI drops. Is this confirmed in your cohort? Could you provide more insight regarding the correlation between PSA levels and MRI accuracy? This could also be done in terms of indeterminate PI-RR scores assigned for example.

2) Since PI-RR is relatively new and the use of MRI for PCa recurrence is not widespread, I suppose that the number of exam presented to describe Readers' experience indicate PI-RADS reads rather than PI-RR. In this light, I would probably remove these and only leave the years of experience in order to avoid possible confusion among readers.

3) The vast majority of the cohort underwent RP. Do the Authors think that this unbalance in the study population could have had an impact on the results? While the Authors mention this in the limitations of the study, I believe this might be a point of discussion worth expanding.

4) If I am not mistaken, the ref standard is PET-CT but two different radiotracers have been considered. Do the Authors think that this might have influenced the results? Choline might be less accurate than PSMA especially at low PSA levels, possibly confirming as true negative what could have been a false negative on MRI. I think it would be interesting to elaborate on this point

Author Response

Open Review

English language and style

( ) English very difficult to understand/incomprehensible
( ) Extensive editing of English language and style required
( ) Moderate English changes required
(x) English language and style are fine/minor spell check required
( ) I don't feel qualified to judge about the English language and style

Yes

Can be improved

Must be improved

Not applicable

Does the introduction provide sufficient background and include all relevant references?

(x)

( )

( )

( )

Are all the cited references relevant to the research?

(x)

( )

( )

( )

Is the research design appropriate?

(x)

( )

( )

( )

Are the methods adequately described?

(x)

( )

( )

( )

Are the results clearly presented?

(x)

( )

( )

( )

Are the conclusions supported by the results?

(x)

( )

( )

( )

Comments and Suggestions for Authors

This is an interesting manuscript dealing with a timely topic. The research question is clear and relevant. The study design is straightforward and properly described. I have only some minor comments:

1) It has been shown that, in the setting of BCR, when PSA level is below 1 ng/ml the accuracy of MRI drops. Is this confirmed in your cohort? Could you provide more insight regarding the correlation between PSA levels and MRI accuracy? This could also be done in terms of indeterminate PI-RR scores assigned for example.

Paragraph has been added in the results and in the discussion. Supplementary table 3 has also been added.

In the group of patients who underwent radical prostatectomy, 83% performed the MRI with a PSA < 1ng/ml, while the remaining 17% with a PSA ≥ 1 ng/ml.

In the “low” PSA group the percentage of patients with a positive outcome (using the above-mentioned composite reference standard) were 42.5% with disease with av-erage size of 4 mm; while in patients with "high" PSA the percentage of positives was 77.8% with a average size of 8 mm.

MpMRI detection performance using a binary evaluation (positive PI-RR score ≥3) showed an higher accuracy in the “high”PSA group ranging between 79%-86% versus 66%-72% in the “low” PSA group. Complete diagnostic performance for each reader in the different PSA setting can be found in supplementary table 3.

2) Since PI-RR is relatively new and the use of MRI for PCa recurrence is not widespread, I suppose that the number of exam presented to describe Readers' experience indicate PI-RADS reads rather than PI-RR. In this light, I would probably remove these and only leave the years of experience in order to avoid possible confusion among readers.

Indeed the number represent the Pi-rads reads, as suggested we deleted the number leaving only the year of expeience.

3) The vast majority of the cohort underwent RP. Do the Authors think that this unbalance in the study population could have had an impact on the results? While the Authors mention this in the limitations of the study, I believe this might be a point of discussion worth expanding.

This limitation in our study has been explained more in details in the limitation paragraph.

4) If I am not mistaken, the ref standard is PET-CT but two different radiotracers have been considered. Do the Authors think that this might have influenced the results? Choline might be less accurate than PSMA especially at low PSA levels, possibly confirming as true negative what could have been a false negative on MRI. I think it would be interesting to elaborate on this point

This has been expanded in the discussion.  Reference 23 has been added

Finally, given the retrospective nature of the study, PET/CT imaging with two different radiotracers has been used to define the reference standard. This might have influenced the obtained results since [18F]F-Fluorocholine PET/CT might have been less accurate than [18F]F-PSMA-1007 PET/CT [25], especially at low PSA levels, possibly confirming as a true negative what could have been a false negative on MRI. Further prospective studies are needed to overcome this limitation. 

Reviewer 2 Report

Introduction:

1.     Lines 60-61 rewrite the sentence, ‘based evaluated’ does not make sense

2.     Same lines – the authors justly suggest that the current limitation of PI-RR is single-centre validation, however, this study has also been performed at a single institution. The next paragraph, however, states the primary aim of the study as evaluating PI-RR reproducibility and interobserver agreement. I suggest that the authors make a smoother transition to the study objective that currently contradicts the unmet need, e.g. by adding an extra sentence in the penultimate paragraph on the lack of reproducibility/inter-reader agreement studies and the clinical significance of this unmet need.

Methods:

1.     It would be helpful to include the precise definition of biochemical recurrence used in this study.

2.     Line 76 – the exclusion criteria ‘were’ – please use past simple when describing the study methods/results

3.     MRI acquisition – it would be helpful for the authors to provide a full mpMRI protocol including TE/TR, slice thickness, gap, etc. for all the sequences

4.     Line 95 – “e” is probably “and”?

5.     Line 131 – examination?

6.     Line 146 – Gwet has proposed an AC1 test in contrast to Cohen’s kappa – what was the rationale for using the former over the latter?

Results:

1.     Please use past simple when describing your results (e.g., the mean age of patients was; the mean disease-free interval was…)

2.     In Table 1, the authors use . to separate the decimal values, whereas elsewhere they use commas (,). For consistency and compliance with the UK/US spelling, please use . throughout the article (e..g, 0.52 instead of 0,52)

3.     In Table 2, please use ≥ sign instead of >=

Discussion:

1.     Line 244 – experience plays

2.     Line 283 – use ‘pitfalls’ instead of ‘drawback’

3.     Line 303 – use ‘limitations’ instead of ‘limits’

4.     I believe the biggest limitation of this study is its single-centre nature (please see my comment 2 for the introduction), which does not address one of the key unmet research needs as highlighted by the authors. Could the authors please comment on this here.

Author Response

Submission Date

13 December 2022

Date of this review

21 Dec 2022 01:02:27

Open Review

English language and style

( ) English very difficult to understand/incomprehensible
( ) Extensive editing of English language and style required
( ) Moderate English changes required
(x) English language and style are fine/minor spell check required
( ) I don't feel qualified to judge about the English language and style

Yes

Can be improved

Must be improved

Not applicable

Does the introduction provide sufficient background and include all relevant references?

( )

(x)

( )

( )

Are all the cited references relevant to the research?

(x)

( )

( )

( )

Is the research design appropriate?

(x)

( )

( )

( )

Are the methods adequately described?

(x)

( )

( )

( )

Are the results clearly presented?

(x)

( )

( )

( )

Are the conclusions supported by the results?

(x)

( )

( )

( )

Comments and Suggestions for Authors

Introduction:

  1. Lines 60-61 rewrite the sentence, ‘based evaluated’ does not make sense

Changed in :

However, this score has been validated only by single-center studies, with cases evaluated by expert radiologists, requiring further validation [6-7].

  1. Same lines – the authors justly suggest that the current limitation of PI-RR is single-centre validation, however, this study has also been performed at a single institution. The next paragraph, however, states the primary aim of the study as evaluating PI-RR reproducibility and interobserver agreement. I suggest that the authors make a smoother transition to the study objective that currently contradicts the unmet need, e.g. by adding an extra sentence in the penultimate paragraph on the lack of reproducibility/inter-reader agreement studies and the clinical significance of this unmet need.

Added an extra sentence in the penultimate paragraph

Methods:

  1. It would be helpful to include the precise definition of biochemical recurrence used in this study.

The definition of biochemical recurrence identifies the cut-off in the value of 0.2 ng/mL after radical prostatectomy, [8]; while after radiotherapy, the “Phoenix” defini-tion used, identifies biochemical recurrence at any value ≥ 2 ng/ml compared to the post-radiotherapy nadir [9].

References 8 and 9 have been added.

  1. Line 76 – the exclusion criteria ‘were’ – please use past simple when describing the study methods/results Changed
  2. MRI acquisition – it would be helpful for the authors to provide a full mpMRI protocol including TE/TR, slice thickness, gap, etc. for all the sequences

Added in the supplementary material

Sequences

TR(ms)

TE(ms)

B value

Slice Thickness (mm)

Scan time(min sec)

Temporal resolution

T2W TSE Ax

5760

110

3(no gap)

T2W TSE Cor e Sag

6970

110

3(no gap)

T1W Ax

575

20

5 (no gap)

DWI-EPI Ax

5800

68

1500

3(no gap)

DWI-EPI Ax

(used for ADC map)

5200

63

50-750-1000

3(no gap)

DCE

4,6

1,76

3(no gap)

2,26 min

9 sec

Supplementary Table 1

Acquisition parameters for multiparametric magnetic resonance imaging of the

prostate.

  1. Line 95 – “e” is probably “and”? Changed
  2. Line 131 – examination? deleted
  3. Line 146 – Gwet has proposed an AC1 test in contrast to Cohen’s kappa – what was the rationale for using the former over the latter?

Gwet’s AC was demonstrated to outperform Kappa and, in particular, it can be considered a paradox-resistant variant of Kappa.

Different studies demonstrated the goodness of Gwet’s AC1 as compared to Kappa: Zhao et al 2022 (https://pubmed.ncbi.nlm.nih.gov/36038846/), Wongpakaran et al 2013 (https://bmcmedresmethodol.biomedcentral.com/articles/10.1186/1471-2288-13-61) and Honda et al 2020 (https://bmcmedresmethodol.biomedcentral.com/articles/10.1186/s12874-019-0887-5).

If reviewer agree I would not change this in the text.

Results:

  1. Please use past simple when describing your results (e.g., the mean age of patients was; the mean disease-free interval was…) Changed
  2. In Table 1, the authors use . to separate the decimal values, whereas elsewhere they use commas (,). For consistency and compliance with the UK/US spelling, please use . throughout the article (e..g, 0.52 instead of 0,52) Changed troughout the article
  3. In Table 2, please use ≥ sign instead of >= Changed

Discussion:

  1. Line 244 – experience plays Changed
  2. Line 283 – use ‘pitfalls’ instead of ‘drawback’ Changed
  3. Line 303 – use ‘limitations’ instead of ‘limits’ Changed
  4. I believe the biggest limitation of this study is its single-centre nature (please see my comment 2 for the introduction), which does not address one of the key unmet research needs as highlighted by the authors. Could the authors please comment on this here.

Added an extra sentence in the penultimate paragraph

A further limitation is the single center design of this study, validation of PI-RR score using MRI images from multiple centers to assess the reproducibility of this score remains an unmet need.

Submission Date

13 December 2022

Date of this review

02 Jan 2023 15:13:06
